# Broadly neutralizing SARS-CoV-2 antibodies through epitope-based selection from convalescent patients

Romain Rouet[1,2,12] ✉, Jake Y. Henry [1,2,12], Matt D. Johansen [3,12], Meghna Sobti[2,4,12], Harikrishnan Balachandran[5,6], David B. Langley[1,2], Gregory J. Walker[5,6,7], Helen Lenthall[1,2], Jennifer Jackson[1,2], Stephanie Ubiparipovic[1,2], Ohan Mazigi[1,2], Peter Schofield[1,2], Deborah L. Burnett [1,2], Simon H. J. Brown [8], Marianne Martinello[5,6], Bernard Hudson[9], Nicole Gilroy[10], Jeffrey J. Post [7], Anthony Kelleher [5,6], Hans-Martin Jäck[11], Christopher C. Goodnow [1,2], Stuart G. Turville [5,6], William D. Rawlinson [5,7], Rowena A. Bull [5,6,13], Alastair G. Stewart [2,4,13], Philip M. Hansbro[7,13] & Daniel Christ [1,2,13] ✉

Emerging variants of concern (VOCs) are threatening to limit the effectiveness of SARS-CoV-2 monoclonal antibodies and vaccines currently used in clinical practice; broadly neutralizing antibodies and strategies for their identification are therefore urgently required. Here we demonstrate that broadly neutralizing antibodies can be isolated from peripheral blood mononuclear cells of convalescent patients using SARS-CoV-2 receptor binding domains carrying epitope-specific mutations. This is exemplified by two human antibodies, GAR05, binding to epitope class 1, and GAR12, binding to a new epitope class 6 (located between class 3 and 5). Both antibodies broadly neutralize VOCs, exceeding the potency of the clinical monoclonal sotrovimab (S309) by orders of magnitude. They also provide prophylactic and therapeutic in vivo protection of female hACE2 mice against viral challenge. Our results indicate that exposure to SARS-CoV-2 induces antibodies that maintain broad neutralization against emerging VOCs using two unique strategies: either by targeting the divergent class 1 epitope in a manner resistant to VOCs (ACE2 mimicry, as illustrated by GAR05 and mAbs P2C-1F11/S2K14); or alternatively, by targeting rare and highly conserved epitopes, such as the new class 6 epitope identified here (as illustrated by GAR12). Our results provide guidance for next generation monoclonal antibody development and vaccine design.

The coronavirus SARS-CoV-2, the causative agent of the global COVID-19 pandemic, has resulted in the death of over 6 million people worldwide[1]. Upon SARS-CoV-2 infection, the human adaptive immune response generates antibodies against the viral spike surface glycoprotein[2]. Most of the neutralizing antibodies bind the spike receptor binding domain (RBD), and in particular class 1 and 2 epitopes within the receptor binding motif (RBM), directly blocking interaction with the human angiotensin converting enzyme receptor 2 (ACE2)[3–6]. Such RBD- and RBM-targeted antibodies are also generated upon vaccination[7], but often lack neutralization potential against emerging

variants of concern (VOCs)[8–10]. More recently, broadly neutralizing antibodies have been identified that bind outside the RBM region[11–15]. Such antibodies generally bind to regions conserved among sarbecoviruses (class 3, 4 and 5 epitopes) and are expected to be more resilient to VOCs. However, it is also evident that such antibodies are rare in most vaccinated individuals and convalescent patients[5,14].

In addition to vaccination, the use of recombinant antibodies has proven successful for therapy and prevention of COVID-19 and several monoclonal antibodies have obtained regulatory approval, and have shown particular promise in immunocompromised individuals and the elderly[16–18]. However, increased resistance to antibody neutralization is being observed for SARS-CoV-2 VOCs, especially for variants carrying mutations within the RBM region (including Beta and Omicron variants)[8–10]. To overcome such resistance, research has focused on the development of broadly neutralizing antibodies, such as Sotrovimab (S309), which targets the class 3 epitope, which is conserved in sarbecoviruses[19]. In addition, antibody cocktails, targeting multiple non-overlapping epitopes on the surface of the RBD have been developed to overcome resistance[20,21].

Here we outline a strategy based on the sorting of convalescent patient B cells in an epitope-specific manner, in combination with structural characterization by X-ray crystallography, cryo-electron microscopy (cryo-EM), live virus neutralization and animal models. Using recombinant RBD proteins carrying epitope-specific mutations, we identified neutralizing antibodies binding to diverse epitope classes. Our approach identified a panel of three non-overlapping antibodies (GAR05, GAR12, and GAR20) binding to epitope classes 1, 6 and 4) that effectively neutralize live virus in vitro and protect K18-hACE2 mice from intranasal challenge. Although these monoclonals had been isolated from convalescent patients infected early in the pandemic (ancestral strain), all three effectively neutralized the immune evasive Delta VOC, with two of the three (GAR05, GAR12) also effectively neutralizing all analyzed Omicron subvariants, highlighting the robustness of the approach. In addition to the previously described class 1-5 epitopes of the RBD, our work identified a new epitope class conserved in sarbecoviruses, that we define as class 6, located between class 3 (exemplified by the antibody S309[19]) and class 5 (exemplified by the antibody S2H97[12]) epitopes. Taken together, our approach identified new broadly neutralizing antibodies from convalescent patients and outlined a new RBD epitope class (class 6), highlighting the potential of the strategy for expanding our understanding of the SARS-CoV-2 antibody response and for future vaccine and therapeutic applications.

## Results
### Epitope-based selection of human memory B cells using mutant SARS-CoV-2 RBDs
We utilized peripheral blood mononuclear cells (PBMCs) from convalescent patients from the COSIN study (New South Wales COVID-19 patient cohort; patients diagnosed by RT-PCR in March 2020 and follow-up samples collected between May and November 2020)[22], at 1- and 4-months post SARS-CoV-2 infection. In order to rapidly identify antibodies binding to different epitopes, we sorted memory B cells based on their capacity to bind to RBD variants carrying epitope class-specific mutations. We initially investigated whether to utilize full spike glycoprotein (trimeric; ancestral strain;[23] randomly biotinylated) or recombinant RBD (single biotin-tag) to select human memory B cells. Using fluorescently-labeled tetramerized spike and tetramerized RBD, we observed a population of CD3$^-$CD19$^+$CD20$^+$CD10$^-$IgD$^-$IgG$^+$ B cells with a high mean of fluorescence intensity (MFI) for both the RBD and spike (albeit lower MFI for spike, Fig. 1a left panel and Supplementary Fig. 1 - Sort 1). Subsequent sorting (Fig. 1a right panels, Supplementary Fig. 1; Sort 2 and Sort 3) focused on the use of tetramerized RBD and a series of SARS-CoV-2 RBD mutants to target different epitope classes. We specifically designed RBD mutants to differentiate antibodies

blocking the ACE2 interactions and antibodies binding outside the ACE2 binding site. For this purpose, we utilized the following mutant SARS-CoV-2 RBDs: Mut1 (T500A/N501A/Y505A), perturbing the ACE2 binding site (targeting class 1 and 2 antibody binders); Mut2 (L455A/F456A), blocking a different surface of the ACE2 binding site (further targeting further class 1 and 2 binders); and Mut3 (K378S), blocking the CR3022 antibody epitope (targeting class 4 binders) (Fig. 1b)[24]. In addition to mutant RBDs, we also utilized SARS-CoV-1 RBD in order to identify broadly neutralizing antibodies (all RBDs were tetramerized with four distinct fluorescent dyes (Supplementary Fig. 1). Having observed RBD/spike cross-specificity (Sort 1), the next strategy (Sort 2) employed SARS-CoV-1, SARS-CoV-2 and Mut1 and Mut3 RBDs, classifying cells as falling into class 1/2, 4 or unknown epitope classes (Fig. 1a and Supplementary Fig. 1). We consolidated these findings using a third sorting strategy (Sort 3) employing WT SARS-CoV-2, Mut1, Mut2 and Mut3 (Fig. 1a and Supplementary Fig. 1 – Sort 3), strengthening delineation of binders that compete with the ACE2 interface.

### Epitope binning and characterization of human monoclonal antibodies
Following the indexed sorting strategies, human antibody variable regions were amplified from isolated B lymphocytes, cloned into human IgG1 expression vectors, and the encoded monoclonal antibodies expressed and purified[25,26]. For an initial screen, we assessed antibody binding by ELISA and found that 80% (16/20) of the antibodies selected bound to WT SARS-CoV-2 RBD (Supplementary Fig. 2a). We then measured the antibody binding affinities to SARS-CoV-2 RBD by BioLayer Interferometry (BLI), which ranged from 290 nM to 200 pM (Fig. 1c and Supplementary Fig. 2b).

To further validate class 1/2 antibodies, we performed BLI competition assays with recombinant human ACE2. Amongst the 16 antibodies screened, we identified seven that competed with ACE2 and thus could be considered as class 1/2 candidates (GAR04, 05, 06, 07, 09, 15 and 20) (Supplementary Fig. 3). All but one of these (GAR04) were capable of binding the E484K RBD mutant, a residue pivotal to the binding mode of many class 2 binders[6], hence we tentatively assigned GAR04 as a class 2 binder, while the other six antibodies were assigned as class 1 binders (Fig. 1c). IGHV3-53, IGHV1-2 and IGHV3-30 germlines have been commonly observed among class 1 and class 2 SARS-CoV-2 binders[27], and we also observed these germlines among several of these selected clones (including GAR01, GAR04, and GAR15; Supplementary Table 3 and Supplementary Table 4).

To further validate class 4 antibodies, we first investigated cross-reactivity with other sarbecovirus RBDs, which are known to be relatively conserved within this epitope (SARS-CoV-1, pangolin CoV and bat RaTG12-CoV)[12,13]. We performed BLI binding experiments with these three sarbecovirus RBDs and identified seven cross-reactive antibodies (GAR01, 03, 11, 13, 14, 16 and 20) (Supplementary Fig. 4). BLI competition assays performed on these cross-reactive antibodies with the class 4 monoclonal antibody CR3022[28,29] (known to bind both SARS-CoV-1 and SARS-CoV-2), confirmed that all, except GAR01 and GAR03, bind to the class 4 epitope (Supplementary Fig. 3). In addition to competing with CR3022, GAR20 also blocked binding of recombinant ACE2 to SARS-CoV-2 RBD, as observed for antibody 3467[14], however this was not observed for the other class 4 antibodies.

The finding that GAR01 and GAR03 are broadly cross-specific and bind to epitopes conserved in SARS-CoV-2, SARS-CoV-1, pangolin CoV and bat RaTG12-CoV (but do not compete with ACE2 nor CR3022) indicate that these two antibodies may be rare binders falling into the class 5 epitope bin. To investigate this hypothesis, we performed BLI competition assays with the class 5 antibody S2H97[12], and confirmed that binding of both antibodies is indeed consistent with the class 5 classification (Supplementary Fig. 3).

Overall, our epitope-binning strategy assigned most of the antibodies to established class 1-5 epitope bins (Fig. 1c)[30]. In addition,

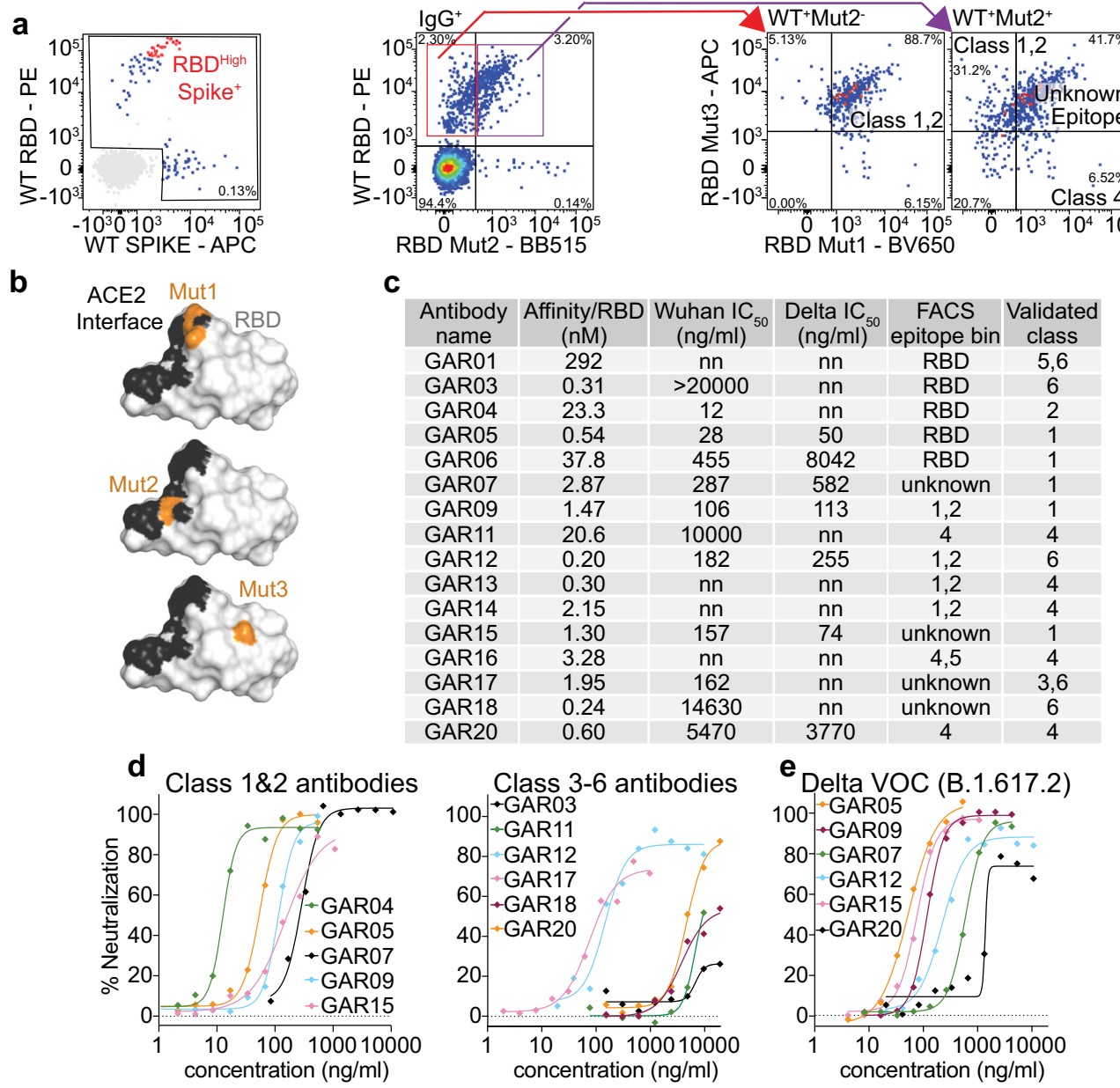

**Fig. 1 | PBMC single cell sorting strategy and antibody characterization.**
**a** Memory B cells were selected from convalescent patient PBMCs by gating on CD3⁻CD19⁺CD20⁺CD10⁻IgD⁻IgG⁺ cells (Supplementary Fig. 1; upper panel) and single cells sorted by binding to fluorescently labeled SARS-CoV-2 RBD or trimeric spike initially (left panel). Red dots represent the B cells for which monoclonal antibodies were amplified and characterized (Supplementary Fig. 1; Sort 1). In addition, using **b** mutant RBD protein (mutations colored in orange), single cells were sorted based on FACS epitope bins (**a**, right panels, Supplementary Fig. 1; Sort 3). **c** Monoclonal antibodies analyzed, showing affinity for SARS-CoV-2 RBD (ancestral strain), neutralization potential and epitope class (initial FACS epitope bin and validation). nn = non neutralizing. In vitro neutralization of live (**d**) ancestral and (**e**) Delta VOC SARS-CoV-2 virus. $n = 4$ technical replicates, data are presented as mean values. Source data are provided as a Source Data file.

antibodies originally sorted into unknown epitope bins were further identified as falling either into epitope class 1/2 (unaffected by T500A/N501A/Y505A and L455A/F456A RBD mutations) or alternately as falling into a new epitope class (class 6, as below).

Next, we evaluated the capacity of these antibodies to neutralize live virus. We observed that 12/16 of the monoclonal antibodies neutralized SARS-CoV-2 ancestral (D614G) strain with $IC_{50}$s ranging from 20 μg/ml to 12 ng/ml (Fig. 1c, d). Several (6/16) antibodies also neutralized the Delta VOC (GAR05, GAR07, GAR09, GAR12, GAR15 and GAR20) (Fig. 1e). Three of these antibodies, GAR05, GAR12 and GAR20, were particularly broad in their specificity and neutralized all analyzed variants of concern, with GAR05 neutralizing with $IC_{50}$s ranging from

115 to 26 ng/ml, and GAR12 and GAR20 neutralizing with $IC_{50}$s ranging from 255 to 128 ng/ml and 12 to 4 μg/ml respectively (Supplementary Fig. 5).

## GAR05 is a broadly neutralizing class 1 antibody

Based on its broad neutralization potential, antibody GAR05 was further characterized by cryo-EM and X-ray crystallography. For cryo-EM studies, GAR05 Fab was incubated with stabilized D614G ancestral trimeric spike ("VFLIP")[31] at a 3:1 molar ratio. The complex was subsequently flash-frozen and examined using cryo-EM (Supplementary Fig. 6 and Supplementary Table 1). The cryo-EM map of GAR05 bound to trimeric spike (Fig. 2a) shows three antibody Fabs bound to the RBD

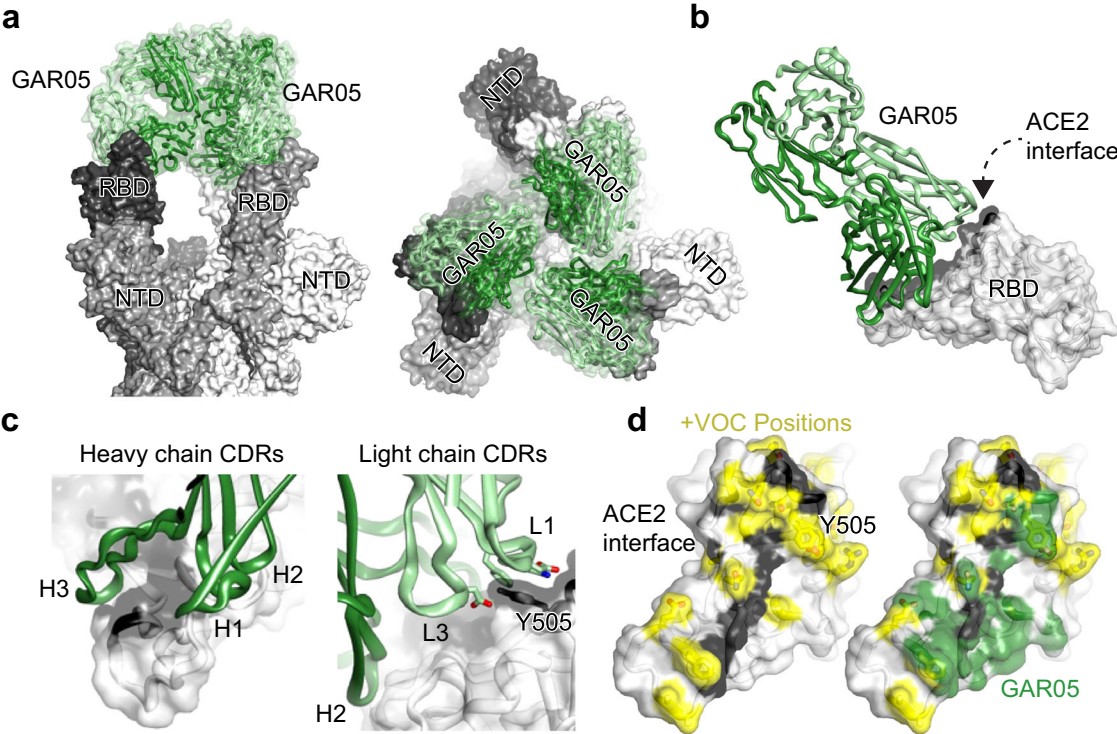

**Fig. 2 | Structural characterization of the broadly neutralizing class 1 antibody GAR05. a** Cryo-EM structure of GAR05 Fab bound to trimeric spike (3.27 Å resolution) showing full antibody Fab occupancy of all the RBDs in the "up" conformation. Two perspectives are shown **b** Structure of GAR05 bound to SARS-CoV-2 RBD based on the X-ray crystal structure, outlining the antibody bound to the ACE2 "saddle" of the RBD (ACE2 interface shaded black). **c** Interaction of the CDR regions of the VH and VL domains with the RBD saddle and the Y505 side chain of the RBD. **d** Comparison of the ACE2 interface on the surface of RBD and the GAR05 epitope, showing high overlap within the ACE2 saddle region. The large, buried surface area of GAR05 (1275 Å²) indeed blankets many key residues identified in VOCs, colored in yellow, yet remarkably still binds all VOCs with high affinity.

domains of the trimeric spike, where all three RBDs are in the up conformation, consistent with its classification as a class 1 antibody (similar to what has been observed for antibody C105[32]). We also solved the crystal structure of GAR05 as a Fab fragment in complex with SARS-CoV-2 RBD to 3.2 Å resolution (Fig. 2b, Supplementary Table 2), essentially confirming the binding mode suggested by the cryo-EM map. GAR05 straddles the middle of the saddle-like RBD surface known to interact with the ACE2 receptor (Fig. 2b). The antibody forms a large RBD-interface (-1275 Å², green surface in Fig. 2d), 815 Å² of which is contributed by the heavy chain. Heavy-chain complementarity determining regions (CDRs) form a cleft bordered on one side by two finger-like loops (CDRs H1 and H2), opposite a thumb-like loop (extended CDR H3) between which the saddle of the RBD (residues 470-492, Fig. 2c) is bound. CDRs L1 and L3 of the light chain also contact the RBD, forming an additional cleft accommodating the side chain of residue Y505, which projects from an adjacent surface of the RBD (Fig. 2c). The large surface area buried at the interface is at the upper end of what is commonly observed for antibody/antigen interactions[33,34], and is likely reflected by the high affinity of GAR05 towards the RBD (540 pM). However, the surface coverage of GAR05 is in close proximity to multiple positions mutated in various Omicron lineages (Fig. 2d). Rationalisation of why tight binding of GAR05 is nevertheless observed can be made by superposition of the GAR05 complex with cryo-EM and crystal structures of an ensemble of Omicron variants (B1.1.529, BA.2, BA.4/5) from a range of biophysical circumstances (RBD down, RBD up, RBD up and complexed by ACE2, see Supplementary Fig. 7). Of the constellation of positions mutated in Omicron VOCs, eight are likely to interface with GAR05. Four of these (S477N, T478K, E484A and F486V) adorn a loop demonstrating considerable flexibility and which forms one end of the saddle bound by ACE2 (Supplementary Fig. 7b). Some positions are mutated to smaller side chains and thus are unlikely to present a steric clash (E484A, F486V, K417N and Y505H; Supplementary Fig. 7, panels b-f). Some positions are mutated to larger side chains albeit presenting in a variety of side chain conformations, some of which might accommodate GAR05 binding (Q493R and N501Y; panel f). Ten hydrogen bonds exist between GAR05 and the RBD (panels g-i). Two of these involve the light chain epitope centered on Y505, both of which will likely be lost by Omicron mutants Y505H and N501Y (panel g). The remaining 8 are heavy-chain centric, 7 involved in contacts with conserved RBD targets (both main-chain and side-chain), and only one will be lost by the E484A mutation (panel i). Hence, the ability of GAR05 to maintain tight binding to Omicron VOCs appears due to combinatorial effects of; coverage of a large epitope surface, the redundancy of heavy and light chains effectively targeting different surfaces (heavy chain targeting the RBD saddle feature, and the light chain targeting the adjacent Y505 feature), the apparent flexibility noted in part of the RBD heavy-chain epitope, the lack of any obvious side-chain mutation likely to present as an unavoidable steric block, and the bulk of hydrogen bonds targeting conserved features.

The binding mode of GAR05 is similar to that of mAb P2C-1F11, which has been described as an ACE2 mimetic due to sharing extensive steric clash volume with ACE2[35]. A primary difference, however, is the higher affinity and extremely long HCDR3 loop of GAR05, which wraps further than P2C-1F11 around the RBD saddle targeted by ACE2 (Supplementary Fig. 8, top panels). A further class I targeting antibody, S2K146[36], has also been described as an ACE2 mimetic as it targets multiple evolutionarily conserved residues utilised by ACE2, and displays broad resistance to VOCs. The overall binding mode of S2K146 resembles that of GAR05, with the notable exception that the orientation of the antibody heavy and light chains are reversed (approximately 180-degree rotation; Supplementary Fig. 8, lower panels).

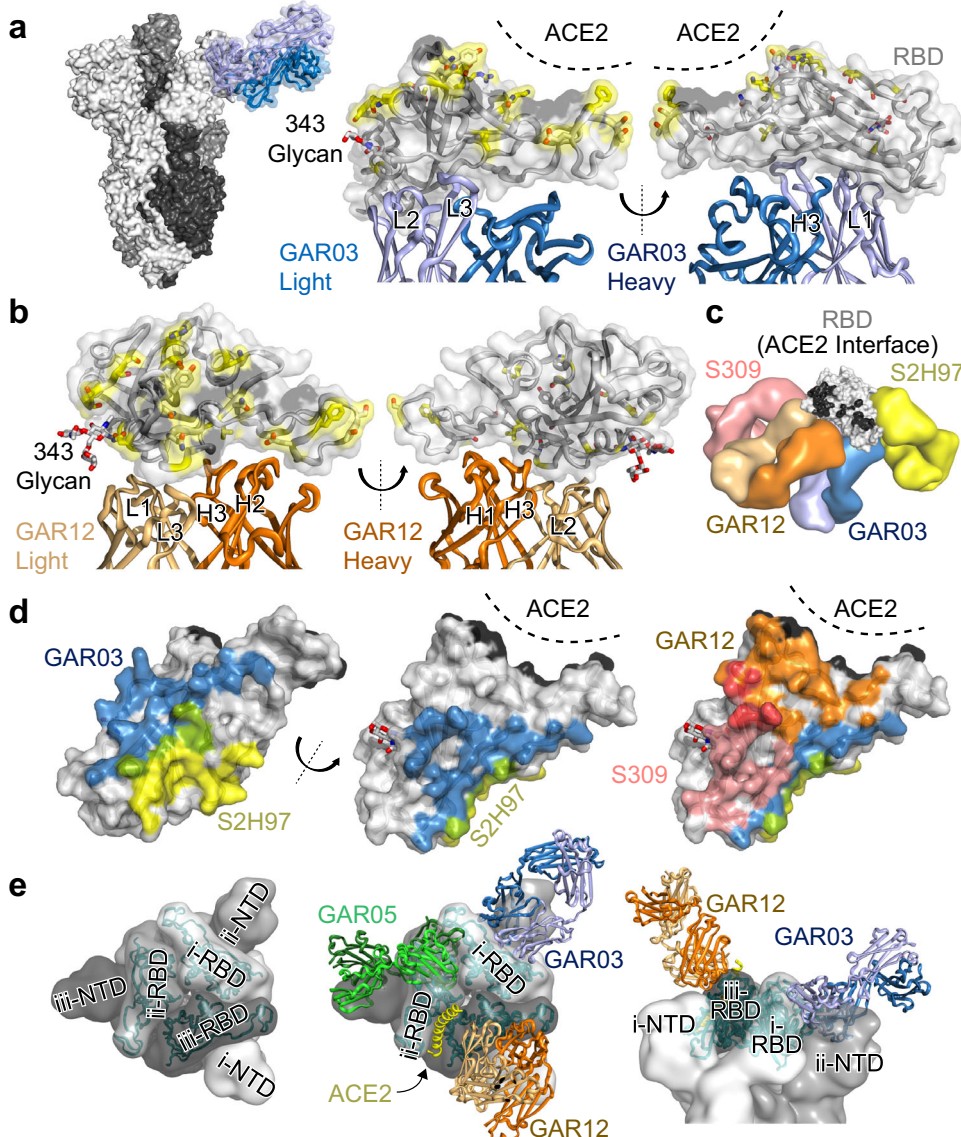

**Fig. 3 | Structural identification of a novel class 6 epitope. a** Cryo-EM structure of spike trimer (three shades of grey surface) complexed with GAR03 (blue cartoon and transparent surface) revealing a single GAR03 antibody Fab bound to trimeric spike (resolution 3.39 Å). Binding displaces the S1 domain (N-terminal domain and RBD) of one of the protomers (disordered and not visible in the cryo-EM map), with the GAR03 Fab binding to the RBD of the adjacent protomer. Middle and right panels show two perspectives of the RBD-GAR03 crystal structure. The binding interface is dominated by the GAR03 light chain (light blue cartoon; CDRs L1, L2, and L3) with more minor contribution from the heavy chain (dark blue; CDR H3). The ACE2 interface of the RBD (grey surface and cartoon) is shaded black, whilst Omicron VOC mutation positions are indicated by yellow sticks and surface, and N343 linked carbohydrate is shown as sticks. **b** Crystal structure of GAR12 (heavy and light chains colored dark and light orange) in complex with RBD (grey cartoon and surface, with ACE2 interface colored black, and Omicron VOCs colored yellow).

Two perspectives are shown. **c** Global juxtaposition of GAR03 (blue surfaces) and GAR12 (orange surfaces) with class 5 representative S2H97 (yellow) and class 3 representative S309 (pink) shown against the RBD surface (grey, and black indicating the RBD-ACE2 interface). **d** RBD-interface comparisons. Left and middle panels including GAR03 (blue) and S2H97 (yellow) and their overlap (green). The right panel additionally includes the GAR12 (orange) and S309 interfaces (pink), and their overlap (red). **e** Spike trimer (three shades of grey surface) presented with all three RBDs in the down conformation (PDB entry 6xm5) as viewed down the approximate 3-fold axis (left and middle panels); docked Fabs (middle panel) GAR03 (blue cartoons), GAR05 (green cartoons), GAR12 (orange cartoons), as well as the primary helix from ACE2 (yellow cartoon). Modelling indicates that only GAR12 is capable of binding to spike with RBD in the down conformation (right panel, viewed from the side).

## Structural identification of a novel class 6 epitope

We next performed structural studies on two broadly cross-reactive antibodies, GAR03 and GAR12, binding conserved epitopes in proximity to that of antibody S2H97[12]. Initial cryo-EM studies using a GAR03-spike mix that had been incubated for 30 min were unsuccessful due to significant aggregation and sample degradation. However, a shorter 1 min incubation of GAR03:spike at a 3:1 ratio improved sample integrity and this preparation was utilized to solve the structure of GAR03 bound to spike glycoprotein (Supplementary Fig. 6 and

Supplementary Table 1). Examination of the cryo-EM map revealed that density could not be observed for the N-terminal domain (NTD) and RBD of one of the three protomers (essentially an S1 domain), with the GAR03 Fab instead occupying the space of the NTD while being bound to the RBD of an adjacent protomer (Fig. 3a). Given that the S2 domain of all three protomers are visible, we hypothesize that binding of GAR03 to an RBD causes the NTD and RBD of an adjacent protomer to be displaced and become flexible, with the averaging methods employed in single particle analysis cryo-EM making their location

unidentifiable. Subsequently, we crystalized the GAR03 Fab in complex with the RBD and solved the structure by X-ray crystallography to 2.75 Å resolution (Fig. 3a, middle and right-hand panels, and Supplementary Table 2), confirming the binding mode suggested by cryoEM, whereby the bulk of the contact interface is mediated by the antibody light chain, and where the binding epitope is well separated from Omicron variant mutation positions (Fig. 3a, positions highlighted in yellow). Competition binding assays suggested that GAR03 binds to an epitope shared with the class 5 antibody S2H97 (Supplementary Fig. 3). Surprisingly, however, mapping of the GAR03 structure onto the S2H97-RBD structure (PDB 7m7w)[12] revealed that, although GAR03 and S2H97 block each other from binding to the RBD, the epitope overlap is minimal (Fig. 3c, d – overlap colored green). More specifically, the GAR03 epitope occupies RBD surface extending from the class 5 epitope exemplified by S2H97 to the class 3 epitope site, exemplified by the monoclonal antibody Sotrovimab (S309, PDB 7r6x[12], Fig. 3c – S2H97 in yellow, S309 in pink, and GAR03 in shades of blue). This epitope is highly conserved among sarbecoviruses, as highlighted by the cross-reactivity of GAR03 with SARS-CoV-2, SARS-CoV-1, Pangolin CoV and Bat RaTG12-CoV RBD. BLI competition experiments revealed that two other effectively neutralizing antibodies, GAR12 and GAR17, competed both with GAR03 and S309 (class 3 epitope) (Supplementary Fig. 9). However, the lack of competition with S2H97 suggests that GAR12 and GAR17 presumably target regions more oriented towards the class 3 epitope site (Fig. 1c). To validate this hypothesis, we solved the crystal structure of the GAR12 Fab in complex with RBD to 2.25 Å resolution (Fig. 3b and Supplementary Table 2). Indeed, GAR12 binds to an epitope on the surface of the RBD that overlaps with S309 and abuts the ACE2 contact surface (Fig. 3b–d, orange cartoon and surfaces). The GAR12 epitope broadly avoids residues mutated in VOCs, thus demonstrating the broad and effective neutralization of this antibody (Fig. 3b). We designate this epitope, shared by GAR03 and GAR12 and spanning a triangular surface between class 5, class 3 and class 1, as class 6 (Fig. 3c).

Superposition of GAR12, GAR03 and GAR05 antibodies onto the trimeric spike protein with the RBDs in the down position indicates that GAR12 is unimpeded and capable of binding to spike in the fully down position (Fig. 3e – orange Fab cartoons). In contrast, neither GAR03 (blue Fab cartoons) nor GAR05 (green Fab cartoons) can access their respective RBD epitopes in the down position due to steric obstruction from either neighboring N-terminal (both) or RBD domains of adjacent protomers (GAR05) within the spike trimer (Fig. 3e).

### Live virus challenge in the K18-hACE2 mouse model

To further validate monoclonal antibodies identified in this study, live virus challenge experiments were carried out using the K18-hACE2 mouse model of SARS-CoV-2 infection in prophylaxis and therapeutic treatments[37,38]. For this purpose, we selected three monoclonal antibodies with broad neutralization of VOCs (Fig. 1c, e, Supplementary Fig. 5) and non-overlapping epitopes (Supplementary Fig. 9), to enable future application as combination therapy. More specifically, we selected GAR05 for class 1, GAR12 for class 6 and GAR20 for class 4.

We initially evaluated the antibodies in a prophylaxis model using live SARS-CoV-2 virus (ancestral). Mice were injected intraperitoneally with 30 mg/kg of monoclonal antibody three days prior to challenge with $1 \times 10^3$ PFU live virus. Mice administered with monoclonal antibodies maintained consistent body weight throughout the challenge compared to mice administered with human IgG1 isotype control (Fig. 4a). Moreover, treatment provided considerable protection from clinical symptoms, with only control mice displaying severe scores (Fig. 4b). Viral titers in lung homogenates from mice treated with monoclonal antibodies were significantly reduced compared to isotype control and were generally below the detection limit (Fig. 4c). Investigation of the bronchoalveolar lavage fluids (BALF) revealed that all three treatment groups had statistically significant reductions in

inflammatory innate immune cells (macrophages and neutrophils), while maintaining undetectable levels of lymphocytes (Supplementary Fig. 10).

In a subsequent study, mice were challenged with ancestral SARS-CoV-2 in a therapeutic setting, with GAR05 administered post-viral challenge at 30 mg/kg. This experiment revealed that, as had been observed in a prophylactic setting, mice treated with GAR05 were fully protected from viral challenge with no measurable weight loss (Fig. 4d), reduction in clinical scores (Fig. 4e) or detectable lung viral titers (Fig. 4f). In a third in vivo setting, GAR05 was used as a prophylactic modality (as above), and mice challenged with the Delta (B.1.617.2) VOC. As had been observed for the initial live virus challenge (ancestral), mice treated with GAR05 were protected from the viral challenge with the Delta VOC: indeed, no measurable reduction of either weight (Fig. 4g), clinical scores (Fig. 4h) could be observed, with undetectable viral titers in the prophylaxis group (Fig. 4i).

Taken together, these findings demonstrate that the monoclonal antibodies developed here, targeting non-overlapping epitopes, broadly and effectively protect human ACE2 mice from SARS-CoV-2 live virus challenge, highlighting their potential for therapeutic applications.

### Broad neutralization of Omicron VOCs

The Omicron VOC and its subvariants have caused worldwide outbreaks, highlighting the importance of broad antibody neutralization and mutational resistance[39,40]. When Omicron BA.1 (B.1.1.529) first emerged in November 2021, we and others had shown that only 1/6 monoclonal antibodies in clinical practice fully maintained activity, namely the broadly neutralizing class 3 monoclonal Sotrovimab (S309)[10,41,42]. To assess the potency of GAR05 and GAR12 against Omicron BA.1 and subvariants, we performed in vitro neutralization assays using live virus. We observed that GAR05 and GAR12 maintained their activity against the analyzed Omicron VOCs (Fig. 5a), with $IC_{50}$ values ranging from 16.17 ng/ml to 337.6 ng/ml, considerably exceeding those of Sotrovimab (S309)[41], particularly for the BA.2 subvariant which is not neutralized by Sotrovimab (Fig. 5b).

### Discussion

The emergence of new SARS-CoV-2 VOCs, including Omicron variants, with increased transmission and immune evasion, has triggered waves of infection throughout the world[43]. Although existing vaccines remain protective against emerging variants (albeit with reduced effectiveness)[44], the vast majority of monoclonal antibodies do not maintain activity against Omicron VOCs[10,45]. We show here that a sorting strategy based on convalescent patient PBMCs, and RBDs bearing class-specific mutations, can select for sarbecovirus cross-specific antibodies, including those binding to epitopes outside the ACE2-binding site[14,46,47]. Such epitope-based sorting strategies using mutant RBDs may provide an efficient method to rapidly identify specific antibodies against future VOCs using variant RBDs or, alternately, those of ancestral strains. This is exemplified by the work outlined here, based on convalescent patients infected with the ancestral strain, which allowed for the identification of antibodies neutralizing a wide range of VOCs. In addition to broadly neutralizing antibodies within each single epitope bin, our strategy allowed for the identification of antibodies in multiple non-overlapping epitope bins, as highlighted by the identification of three non-competing antibodies (GAR05, GAR12 and GAR20) that protect K18-hACE2 mice from live SARS-CoV-2 challenge. This suggests possible future use for combination therapy to increase effectiveness and prevent mutational escape. Using cryo-EM and crystallography, we define a new and highly conserved class 6 RBD epitope on the surface of SARS-CoV-2 RBD, located between class 3 and class 5 epitopes. While the class 6 antibodies identified here do not directly occlude binding to the cellular ACE2 receptor, cryo-EM experiments performed on GAR03 indicated that

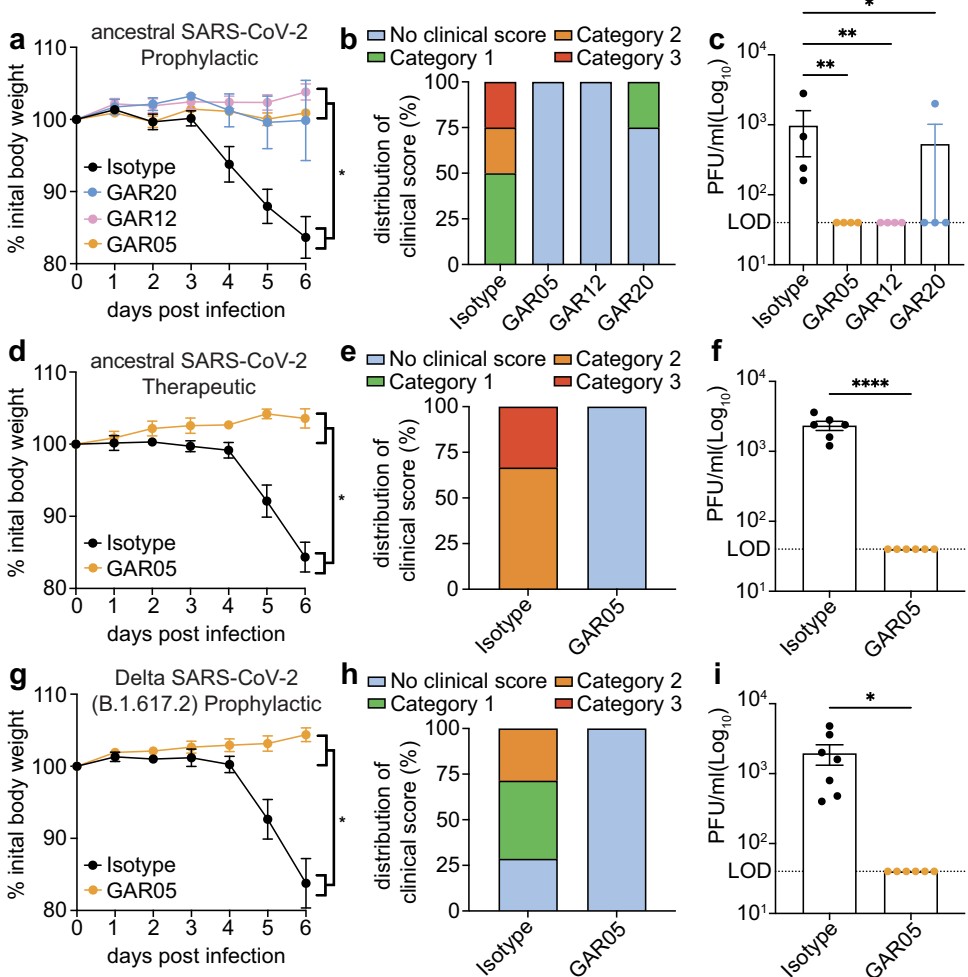

**Fig. 4 | Protection of K18-hACE2 mice from live viral challenge.** Animals were injected interperitoneally with 30 mg/kg of GAR05, GAR12, GAR20 or human IgG1 isotype control 3 days prior to viral challenge (-3 dpi) for prophylactic studies (**a–c**, **g–i**) or +1 dpi post challenge for therapeutic studies (**d–f**). Mice were challenged (d0) with $1 \times 10^3$ PFU of intranasal ancestral SARS-CoV-2 or Delta VOC (B.1.167.2) virus and monitored for 6 days, and euthanized (d6) for tissue collection. Measurements of weight (percentage loss from initial weight, **a** [$n = 4$ mice per group measured on 6 consecutive days. 2-way ANOVA with Tukey's multiple comparisons test. *$p < 0.05$], **d** [$n = 6$ mice per group measured on 6 consecutive days. A single isotype control mouse was removed from the study on day 5 as it

reached a humane endpoint. Mixed-effects analysis with Šídák's multiple comparisons test. *$p = 0.0149$], **g** [$n = 7$ istoype control mice and $n = 6$ GAR05 treated mice measured on 6 consecutive days. 2-way ANOVA with Tukey's multiple comparisons test. **$p = 0.0049$]), clinical score (weight loss, eye closure, appearance of fur and posture, and respiration, **b/e/h**), and viral titers (lung homogenates; plaque assay, **c** [$n = 4$ mice per group. Kruskal-Wallis test of non-parametric data with Dunn's multiple comparisons test. *$p < 0.05$, **$p < 0.01$, **f** [$n = 6$ mice per group. Two-tailed unpaired t-test. ****$p < 0.0001$, **i** [$n = 7$ isotype control mice and $n = 6$ GAR05 treated mice. Two-tailed unpaired t-test. *$p = 0.0179$]) shown. Data are presented as mean values +/- SEM. Source data are provided as a Source Data file.

rapid disruption of the spike structure upon incubation with antibody, suggesting trimer dissociation as a dominant mode of action (similar to what has been observed for the non-ACE2-blocking antibodies CR3022 (a class 4 binder) S2H97 (a class 5 binder) and COV2-3434 (targeting the NTD)[12,48–50]. This conclusion is further supported by a recently reported antibody, 35B5[51], which contacts the class 6 epitope through unconventional framework interactions (involving outer edges of HCDR2, HCDR3, and framework regions FRH1 and FRH3 – rather than the canonical CDR-mediated interactions reported here – Supplementary Fig. 11). Intriguingly, despite its dramatically different binding mode, 35B5 has also been shown to rapidly disrupt the spike trimer, most likely due to steric issues involving NTD proximity[51], further supporting the notion that interactions with the class 6 epitope result in general spike dissociation, independent of specific interaction details. Our results also suggest that exposure to the original ancestral SARS-CoV-2 RBD (through infection or immunization) induces antibodies in patients that maintain robust and broad neutralization potential against emerging VOCs. This is achieved by either targeting

the class 1/2 epitope in a manner resistant to VOCs (as illustrated by GAR05) or alternatively by targeting rare and highly conserved epitopes, such as the novel class 6 epitope described here (as illustrated by GAR12), providing important guidance for vaccine design. We conclude that epitope-based selection of SARS-CoV-2 neutralizing antibodies from convalescent patients enables the identification of promising new antibodies and epitopes against emerging variants of concern.

## Methods

### Antigen production and purification

DNA encoding SARS-Cov-2 RBD (residues 319-541) was gene synthesized (Genscript) and cloned into the pCEP4 mammalian expression vector (Invitrogen) encoding a N-terminal IgG leader sequence and C-terminal Avi- and His-tag. RDB mutants were generated by site-directed mutagenesis and splice-overlapping PCR. Plasmids were transfected into Expi293 cells (Thermo Scientific) according to the manufacturer's protocol and the protein expressed for 7 days at 37 °C, 8% $CO_2$. Cell cultures

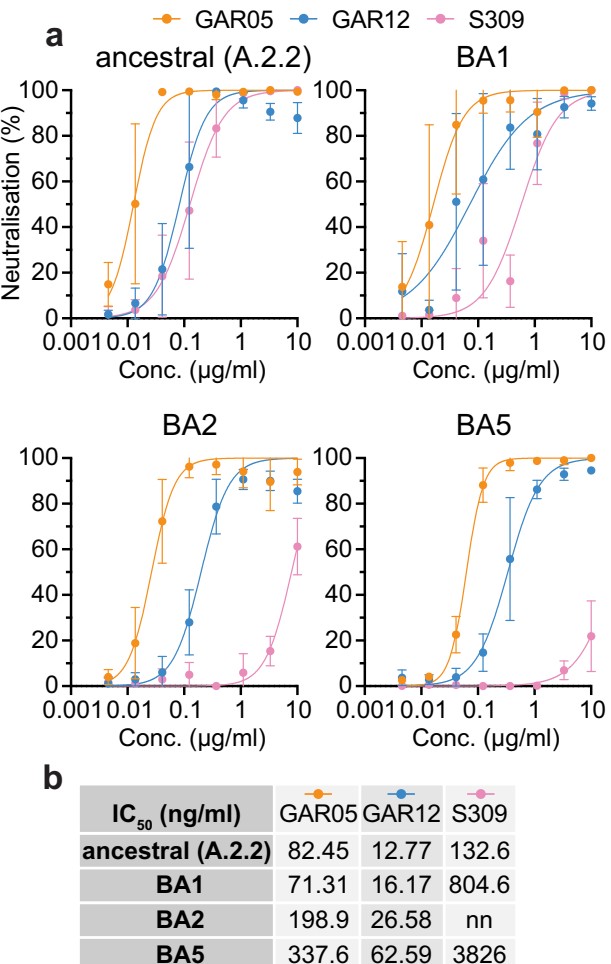

**b**

| IC$_{50}$ (ng/ml) | GAR05 | GAR12 | S309 |
|---|---|---|---|
| ancestral (A.2.2) | 82.45 | 12.77 | 132.6 |
| BA1 | 71.31 | 16.17 | 804.6 |
| BA2 | 198.9 | 26.58 | nn |
| BA5 | 337.6 | 62.59 | 3826 |

**Fig. 5 | Neutralization of the Omicron VOCs. a** SARS-CoV-2 neutralization (live virus), ancestral (A.2.2), Omicron VOCs BA1, BA2, BA5 (*n* = 4 technical replicates). **b** IC$_{50}$ values (ng/ml) reported for S309, GAR05 and GAR12 on ancestral and Omicron VOCs. nn = non-neutralizing. Data are presented as mean values +/- SD for measurements performed over a range of concentrations. Source data are provided as a Source Data file.

were clarified by centrifugation, dialyzed with PBS and the protein captured with Talon resin (Thermo Scientific). The RBDs were eluted with 150 mM imidazole in PBS, dialyzed with PBS and the purity assessed by visualization on SDS-PAGE. The protein concentration was determined by measuring the Abs$_{280nm}$. For biotinylation of the Avi-tag, the proteins were buffer exchanged into 20 mM HEPES pH 7.5 + 50 mM NaCl using Zeba Spin desalting columns (Thermo Scientific). Then 10 mM ATP, 10 mM MgCl$_2$, 150 μM D-biotin and 3 μM BirA enzyme (purified in-house) in 50 mM bicine buffer pH 8.3 was added to the protein and incubated for 1 h at 30 °C. The biotinylated protein was finally buffer exchanged back into PBS and the concentration determined by measuring the Abs$_{280nm}$. The plasmid encoding the ancestral SARS-CoV-2 spike protein with C-terminal trimerization domain and His tag was kindly provided by the Krammer lab (BEI Resources). The plasmid encoding the D614G VFLIP spike protein with a C-terminal trimerization domain and streptavidin tag was kindly provided by the Saphire lab (La Jolla Institute of Immunology). The spike plasmids were transfected into ExpiCHO and the protein expressed for 7 days at 32 °C, 8% CO$_2$. After expression, culture media were clarified by centrifugation and dialysed against 50 mM Tris pH 7.5 + 150 mM NaCl, and the ancestral spike protein was purified using Talon Resin (Thermo Scientific) while the D614G VFLIP spike was purified on a StrepTrap XT column (GE Healthcare), according to the manufacturer's protocols. The spike

proteins were further purified on a Superose 6 gel filtration column (GE Healthcare) in 25 mM Tris pH 7.5 + 150 mM NaCl using an AKTA Pure FPLC instrument (GE Healthcare) to isolate the trimeric protein (and remove S2 pre-fusion protein from the ancestral spike). Finally, the fractions containing the protein were concentrated using Amicon concentrators (Merck), the protein concentration determined by measuring Abs$_{280nm}$ and purity assessed by visualization on SDS-PAGE gel. The spike protein was biotinylated using EZ-Link NHS-PEG4-Biotin reagent (Thermo Scientific) as follows. Purified spike protein was buffer exchanged into PBS using equilibrated Zeba Spin columns. The protein concentration was determined and the spike biotinylated with EZ-Link NHS-PEG4-Biotin at a 10:1 biotin-to-protein molar ratio for 30 min at room temperature. Free biotin was removed from the spike by repeating the buffer exchange step in a second Zeba Spin column equilibrated with 25 mM Tris pH 7.5 + 150 mM NaCl.

### Human PBMC selection
Patient PBMC samples were accessed via the COSIN Study (New South Wales COVID-19 patient cohort, NCT04383652[52]). The study was approved by the Human Research Ethics Committees of the Northern Sydney Local Health District and the University of New South Wales, NSW Australia (ETH00520) and was conducted according to the Declaration of Helsinki and International Conference on Harmonization Good Clinical Practice (ICH/GCP) guidelines and local regulatory requirements. Written informed consent was obtained from all participants before study procedures. PBMC samples were taken from 5 patients (4 male, 1 female), age ranges from 59-72. Patient samples were selected based on high SARS-CoV-2 neutralizing sera titres previously reported in gender and age matched studies[2,53].

### Human B-cell sorting
SARS-CoV-2 specific memory B cells were characterized and sorted via flow cytometric tetramer assay essentially as previously described:[2] biotinylated SARS-CoV-2 RBD (including mutant variants), spike and SARS-CoV-1 RBD were incubated with Streptavidin-APC (BD), Streptavidin-PE (BD), Streptavidin-BB515 (BD) or BV650-PE (BD) at a 4:1 molar ratio. Streptavidin conjugated fluorophores were titrated onto biotinylated antigen in 1/10$^{th}$ volume increments for 10 min per increment at 4 °C.

Cryopreserved PBMCs from five convalescent patients were thawed and suspended in pre-warmed RPMI-1640 media containing 10% FBS (sigma), 2 mM L-glutamine, 50 IU/ml penicillin and 50 μg/ml streptomycin. A maximum of 1 x 10$^7$ cells were resuspended in Fixable Viability Stain 700 (BD) diluted to 1:1000 in PBS and incubated at 4 °C for 20 min. Cells were washed twice with FACS buffer (PBS + 0.1% BSA) and incubated with Human Fc block (5 μl per 2 x 10$^6$ cells; BD) at room temperature for 10 min. Cells were then washed twice with FACS buffer and resuspended in RBD or spike tetramers at 1 μg/ml (per tetramer) and incubated at 4 °C for 30 min and washed twice more with FACS buffer. Cells were finally suspended in a cell surface staining mix containing (per test): 50 μl brilliant staining buffer (BD), 5 μl anti-human CD21 BV421, 5 μl anti-human IgD BV510, 5 μl anti-human CD19 BV711, 5 μl anti-human CD20 APC-H7, 8 μl anti-human IgG BV786, 2 μl anti-human CD27 PE-CF594, 2 μl anti-human CD38 PE-Cy7 and 0.5 μl anti-human CD3 BB700 (all antibodies sourced from BD). Surface staining incubation was performed at 4 °C for 30 min, washed twice and resuspended in FACS buffer for sorting. Immunophenotyping and index sorting was performed on a BD FACSAria III (FACS Diva v6.1.3 software – Becton Dickinson) where a minimum of 300,000 events were acquired for each sample. Data analysis was performed using FlowJo version 10.7.1 (Becton Dickinson).

### Monoclonal antibody production and purification
Sorted single B cells were collected into 96-well PCR plates containing 2 μL of Triton X-100 (0.1%), dNTPs (5 nmol), oligo-dT primer (2.5 pmol)

and RNAse inhibitor (1U – Clontech). cDNA generation and amplification was performed following the SmartSeq2 approach[26]. Briefly, RT mix (100U SuperScript II reverse transcriptase, 10U RNAse inhibitor, Superscript II first stand buffer (1X), 5 mM DTT, 1 M Betaine, 6 mM MgCl$_2$, 100uM TSO in nuclease-free water) was added to samples and reverse transcription was performed using an initial 90 min cycle of 42 °C followed by 10 cycles of alternating 2 min cycles of 50 °C and 42 °C. PCR preamplification was then performed on 10 µl of first-strand reaction by addition of PCR mix (KAPA HiFi HotStart Readymix (1x) with 0.1 µM IS PCR Primers in nuclease-free water) and PCR was performed using 67 °C annealing temperature and 6 min extension at 72 °C for 19 cycles. DNA encoding the antibody variable domains was amplified by nested-PCR following the Tiller protocol[25]. Briefly, in a 20 µl reaction volume, 1.5 µl of cDNA, 0.3 µl of dNTP mix (NEB), 2 µl of Taq HotStart buffer (NEB), 0.25 µl of Taq HotStart DNA polymerase (NEB) and 0.3 µl of forward and reverse primer at 10 µM was added. For VH, the primers consisted of a mix of the 4 VH forward (mixed equimolar concentration) and the CH1 Cγ reverse; for Vkappa, the primers consisted of a mix of the 3 Vk forward primers (mixed equimolar concentration) and the Ck 494 reverse; for Vlambda, the primers consisted of the 7 Vl forward (mixed equimolar concentration) and the Cl reverse, all detailed in Tiller et al. A VH, Vkappa and Vlambda PCR was performed for each cDNA, using 58 °C annealing temperature and 1 min extension at 68 °C for 50 cycles. For the second PCR, we used 5 µl of unpurified PCR product, 0.3 µl of dNTP mix (NEB), 2 µl of Taq HotStart buffer (NEB), 0.25 µl of Taq HotStart DNA polymerase (NEB) and 0.3 µl of forward and reverse primer at 10 µM in a 20 µl reaction. The forward primers for VH, Vkappa and Vlambda were identical as for the 1st PCR, with the exception of a 5′ flanking SapI restriction site. The reverse primers consisted of a mix of the 3 JH segments primers, a mix of the 4 Jkappa segments primers, or a mix of 4 Jlambda segments primers, all flanked with a 3′ SapI restriction site, as detailed in Tiller et al. The PCR reaction followed the same settings as the first PCR. 2 µl of PCR product was ran on a 2% agarose gel and if a band was present around 400 bp for the VH and a VL (kappa or lambda), then the PCR products were purified and digested with SapI restriction enzyme. The VH and VL products were subsequently cloned into in-house human IgG1 and kappa/lambda pCEP4 vectors (containing a stuffer fragment in place of the variable domain, flanked by a 5′- and 3′-SapI restriction site). After validation by Sanger sequencing, plasmids were transfected into ExpiCHO (Thermo Scientific) according to the manufacturer's protocol (1 µg DNA/ml of cells; 2:1 ratio of heavy chain to light chain; following the maximum titer protocol). After 14 days, cell culture media was clarified by centrifugation and the IgG captured using Protein G resin (Genscript). The IgG were eluted from the Protein G resin using 100 mM glycine pH 3.0, dialyzed with PBS and the purity assessed by visualization on SDS-PAGE. For Fab production, the DNA encoding the VH domain was cloned into an in-house human CH1 pCEP4 vectors with a C-terminus His tag. The production was performed in ExpiCHO as for IgG. After 14 days, the cell culture media was clarified by centrifugation, dialyzed against PBS and the Fab captured using Talon resin. The Fab was eluted with 150 mM imidazole in PBS, dialyzed with PBS and the purity assessed by visualization on SDS-PAGE. ACE2 fused to human IgG1 Fc domain was gene synthesized (Genscript) and cloned into pCEP4. Expression was carried out in Expi293 cells for 7 days at 37 °C, 8% CO$_2$. After expression, the ACE2 Fc fusion protein was purified by Protein G resin as for human monoclonal antibody. Antibody sequences were read and designed using Snapgene v6.1.1 (GSL Biotech LLC) software.

## Monoclonal antibody ELISA

Maxisorp plates were coated with SARS-CoV-2 RBD at 2 µg/ml in carbonate coating buffer overnight at 4 °C. The following day, plates were washed twice with PBS and blocked in 4% milk in PBS-T (0.1% Tween 20) for 2 h. Plates were washed 3x with PBS-T and incubated with the monoclonal antibodies diluted to 100 µg/ml in PBS for 1 h. Plates were then washed 3x with PBS-T and incubated with HRP-conjugated goat Anti-human IgG antibody (Jackson ImmunoResearch) (diluted 1:5000 in PBS-T) for 1 h. Plates were finally washed 2x with PBS-T and 1x with PBS and incubated with TMB substrate (Perkin Elmer) until the reaction was quenched with HCl. Plates were read at Abs 450 nm (ClarioStar – BMG Labtech).

## Monoclonal antibody affinity measurements and competition assays by biolayer interferometry (BLI)

Purified monoclonal antibodies were buffer exchanged into PBS using equilibrated Zeba Spin columns. The protein concentration was determined and the antibodies biotinylated by incubating for 30 min at room temperature with EZ-Link NHS-PEG4-Biotinylation reagent (Thermo Scientific) at a 10:1 biotin-to-protein ratio. Free biotin was removed by repeating the buffer exchange step in a second Zeba Spin column equilibrated with PBS. Affinity of interactions between biotinylated antibodies and purified soluble RBD proteins were measured by Biolayer Interferometry (BLItz, ForteBio). Streptavidin biosensors were rehydrated in PBS containing 0.1% BSA for 1 h at room temperature. Biotinylated antibody was loaded onto the sensors "on-line" using an advanced kinetics protocol, and global fits were obtained for the binding kinetics by running associations and dissociations of RBD proteins at a suitable range of molar concentrations (2-fold serial dilution ranging from 800 nM to 50 nM). The global dissociation constant ($K_D$) for each 1:1 binding interaction was determined using the Blitz Pro v1.2.1.3 (Forte Bio) software. For competition assays, RBD at 400 nM was pre-incubated with competitor (ACE2 Fc or unconjugated antibody) at 1 µM for 10 min at room temperature.

## SARS-CoV2 neutralization assay

Serial 2-fold dilutions of test monoclonal antibody were prepared, and neutralization assay was performed in Vero E6 cells (HEK293T cells over-expressing the human ACE2 receptor for the Delta VOC), as previously described[54].

## X-ray crystallography

Antibody Fab fragments were purified further via size-exclusion chromatography using an S200 increase 10/30 column (GE Healthcare) equilibrated with 25 mM Tris (pH 8.0), 200 mM NaCl. C-terminally His-tagged SARS-CoV-2 RBD (residues 333-528) was expressed and purified as for the other RBDs. Fab and RBD were stoichiometrically combined and concentrated to approximately 5 mg/ml. Solutions were combined with an equal volume (2 µL) of well solution in a vapor diffusion hanging drop crystallization format. For the RBD-GAR05 complex, the well solution comprised 100 mM citrate pH 4.25, 500 mM LiCl and 13% (w/v) PEG 6000. For the RBD-GAR03 complex, the well solution comprised 100 mM of the MMT buffer system (pH 6.6) (Molecular Dimensions; comprising malic acid, 2-ethanesulfonic acid and trisaminomethane), and 19% (w/v) PEG6K. To facilitate crystallization this complex also included an additional antibody Fab (10G4) which binds to a different epitope of the RBD (to be described elsewhere; Mazigi et al.). For the RDB-GAR12 complex the well solution comprised 100 mM ammonium citrate (pH 5.5), 20% (w/v) PEG3350. Crystals were flash frozen in liquid nitrogen with no cryoprotection regime. Diffraction data were collected at the Australian Synchrotron on beamline MX2 using a Dectris Eiger X16M detector. For all crystals a 360° sweep of data were deconvoluted into 3600 x 0.1° oscillation images which were indexed and integrated by XDS (version 0.6.5.2)[55]. Space groups were determined with Pointless (version 1.12.12)[56] and scaling and merging performed with Aimless (version 0.7.7)[57], both components of CCP4[58]. Structures were determined by molecular replacement using Phaser (version 2.8.3)[59], using generic human Fab fragments (containing kappa light chain for Fabs GAR05, GAR12 and 10G4, and containing a lambda light chain for

GAR03) and SARS-CoV-2 RBD as search models. All structures contained one complex (1:1, or 1:1:1 for GAR03) in the asymmetric unit. Structure refinement was performed using Refmac (version 5.8.0267)[60] then phenix.refine (version 1.11.1-2775)[61] between rounds of manual inspection and model adjustment using Coot (version 0.9.6)[62]. The electron density for many parts of the GAR05 model, in particular, was poorly defined, commensurate with the low resolution and very high B-factors, hence refinement employed a combination of reference model and secondary structure restraints, however some of the best density exists at the interface between the antibody and the RBD, providing confidence in the Fab-RBD juxtaposition. Model validation was performed using the Molprobity server[63]. Diffraction data and model refinement statistics are shown in Supplementary Table 2.

## Cryo-electron microscopy

For sample preparation, D614G VFLIP spike trimer and Fab (molar ratio of 1:3) was incubated at room temperature for either 30 min (GAR05) or 1 min (GAR03) before application to holey gold grids and freezing. 3.5 μl of each sample was applied to 1.2/1.3 UltrAuFoil grids (Quantifoil) which had been glow-discharged for 1 min at 15 mA in an EasiGlow (Pelco). Plunge freezing was performed using a Vitrobot Mark IV (Thermo Scientific) with 0 blot force, 4 s blot time and 100% humidity at 22 °C. Grids were initially screened on a Talos Arctica Electron Microscope (Thermo Scientific) and assessed for particle concentration, integrity, and ice thickness. For final data collection, grids were transferred to a Titan Krios Electron Microscope (Thermo Scientific) operating at 300 kV equipped with a BioQuantum K2 (Gatan) with the slit set to 20 ev. Movies were recorded using EPU with a calibrated pixel size of 1.08 Å, a total dose of 50 electrons spread over 50 frames and a total exposure time of 5 s. All processing was performed in cryoSPARC[64]. Initial particles were picked using the "blob" protocol, and these 2D classified to create templates to pick the datasets. Extracted particles were subjected to multiple rounds of 2D classification, ab initio reconstruction and heterogenous refinement to sort the particles into discrete structures (Supplementary Fig. 6). In the case of the D614G VFLIP spike trimer bound to GAR03, focussed refinement was utilised on the region corresponding to GAR03:RBD in order to increase the detail in this area (Supplementary Fig. 6). An atomic model for GAR03 Fv was generated using ABodyBuilder[65], the constant domains CH1 and lambda CL added using PDB ID 7m3i[66], and fitted into the focussed map. This model was combined with an atomic model for the entire spike complex (PDB ID 6xlu[67]), introducing the D614G VFLIP mutations and removing regions with little to no density. This model was then used to create a combined map (with the focussed and unfocussed maps) in Phenix[68] and the model was fitted/refined using Coot[62], ISOLDE[69] and Phenix. Supplementary Fig. 6 provides a summary of the data collection and refinement statistics.

## Animal experiments

All mouse experiments were performed according to ethical guidelines as set out by the Sydney Local Health District (SLHD) Animal Ethics and Welfare Committee, which adhere to the Australian Code for the Care and Use of Animals for Scientific Purposes (2013) as set out by the National Health and Medical Research Council of Australia. SARS-CoV-2 mouse infection experiments were approved by the SLHD Institutional Biosafety Committee. For ancestral SARS-CoV-2 prophylaxis assays, 4 female hemizygous K18-hACE2 mice (B6.Cg-Tg(K18-hACE2)2Prlmn/J, stock Nb. 034860, Jackson Lab, 6–8 weeks old) were injected on day -3 (prior to intranasal viral challenge with 1 x 10³ PFU of ancestral SARS-CoV-2 (Isolate AUS/VIC01/2020)) intraperitoneally with 30 mg/kg for each individual antibody (GAR05, GAR12, GAR20 and a human IgG1 isotype control (Bio X Cell)) at 3 mg/ml in PBS in a 200 μl volume. For ancestral SARS-CoV-2 therapeutic assays, 6 female hemizygous K18-hACE2 mice (B6.Cg-Tg(K18-hACE2)2Prlmn/J, stock Nb. 034860, Jackson Lab, 6–8 weeks old) were injected on day 1

(post intranasal viral challenge with 1 x 10³ PFU of ancestral SARS-CoV-2 (Isolate AUS/VIC01/2020)) intraperitoneally with 30 mg/kg GAR05 or a human IgG1 isotype control (Bio X Cell)) at 3 mg/ml in PBS in a 200 μl volume. For Delta SARS-CoV-2 prophylaxis assays, 6 female hemizygous K18-hACE2 mice (B6.Cg-Tg(K18-hACE2)2Prlmn/J, stock Nb. 034860, Jackson Lab, 6–8 weeks old) were injected on day -3 (prior to intranasal viral challenge with 1 x 10³ PFU of Delta SARS-CoV-2 (B1.617.2)) intraperitoneally with 30 mg/kg GAR05 and a human IgG1 isotype control (Bio X Cell)) at 3 mg/ml in PBS in a 200 μl volume. On day 0, treated mice were moved to the PC3/BSL3 lab and intranasally inoculated with 1 x 10³ PFU of ancestral SARS-CoV-2 (Isolate AUS/VIC01/2020) or Delta SARS-CoV-2 (B1.617.2) in a 30 μl volume as previously described[38]. Mice were weighed, monitored and scored for clinical symptoms (weight loss, eye closure, appearance of fur and posture, and respiration) daily and any mice that reached the ethical endpoints were humanely euthanised. On day 6, all mice were humanely euthanised. Multi-lobe lungs were tied off and BALF was collected from the single lobe via lung lavage with 1 mL HANKS solution using a blunted 19-gauge needle inserted into the trachea. BALF was centrifuged (300 g, 4 °C, 7 min), and supernatants collected and snap frozen. Cell pellets were treated with 200 μL Red Blood Cell Lysis Buffer (Thermo Scientific) for 5 min, followed by addition of 700 μL HANKS solution to inactivate the reaction and then centrifuged again. Cell pellets were resuspended in 160 μL HANKS solution and enumerated using a haemocytometer (Sigma-Aldrich). Differential cell BALF counts were performed by loading 70 μl of the BALF cell pellet into a cytospin funnel and centrifuging (300 × g, 7 min) and were left to air-dry overnight. Following drying, slides were stained using the Quick Dip stain kit (POCD Scientific) as per manufacturer's instructions. Multi-lobe lungs were collected and snap frozen on dry ice. Lung homogenates were prepared fresh, with the third multi-lobe lungs placed into a gentleMACS C-tube (Miltenyi Biotec) containing 2 mL HANKS solution. Tissue was homogenised using a gentleMACS tissue homogeniser, after which homogenates were centrifuged (300 × g, 7 min) to pellet cells, followed by collection of supernatants for plaque assays. The significance of differences between experimental groups was evaluated by one-way analysis of variance (ANOVA), with pairwise comparison of multi-grouped data sets achieved using Tukey's or Dunnett's post-hoc test. Differences were considered statistically significant when $p \leq 0.05$. All statistical analyses performed using Prism v9.4.0 (Graphpad Software).

## Reporting summary

Further information on research design is available in the Nature Portfolio Reporting Summary linked to this article.

## Data availability

The crystallography data generated in this study have been deposited in the PDB database under accession codes 7t72, 8dxu and 8dxt. The Cryo-EM data generated in this study have been deposited in the PDB database under accession code 7t5o and the EMDB database under entries 25699 and 25700. Source data used for graphs are provided with this paper. Source data are provided with this paper.

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

## Acknowledgements

R.R. acknowledges funding from the Australian Research Council (ACR DE190100985) and from the Garvan Institute of Medical Research (catalytic COVID grant). D.C. acknowledges funding from the Medical Research Future Fund (MRFF MRF2001739) and from the National Health and Medical Research Council (NHMRC APP1157744 and APP1113904). A.G.S. acknowledges funding from the NHMRC (APP1159347). R.A.B. acknowledges funding from the Snow Foundation. P.M.H. acknowledges funding from the NHMRC (APP1175134), MRFF (MRF2007221), NSW RNA Production Network, the Rainbow Foundation and UT. H.M.J. acknowledges grants TRR130 from Deutsche Forschungsgemeinschaft (DFG), grants 01KI2043 and COVIM (NaFoUniMedCovid19; FKZ: 01KX2021) from the Bundesministerium für Bildung und Forschung (BMBF), from the Kastner Foundation and grants from the Bayerische Forschungsstiftung and the Bavarian Ministry of Art and Science. Part of this work was performed on the MX2 beamline at the Australian Synchrotron, part of ANSTO. The authors would like to thank the study participants for their contribution to the research. They would like to acknowledge members of the study group: Protocol Steering Committee – Andrew Lloyd (UNSW), John Kaldor (UNSW), Greg Dore (UNSW), Tania Sorrell (Marie Bashir Institute), William Rawlinson (NSWHP), Jeffrey Post (POWH), Bernard Hudson (RNSH), Dominic Dwyer (NSWHP), Adam Bartlett (SCH), Sarah Sasson (UNSW) Nick Di Girolamo (UNSW) and Daniel Lemberg (SCHN). Coordinating Centre - Marianne Byrne (Clinical Trials Manager), Mohammed Hammoud (Post-Doctoral Fellow and Data Manager), Andrew Lloyd (Investigator) and Roshana Sultan (Study co-ordinator). Site Principal Investigators – Jeffrey Post (POWH), Michael Mina (Northern Beaches Hospital), Bernard Hudson (RNSH), Nicky Gilroy (Westmead Hospital), Pam Konecny (St George Hospital), Golo Ahlenstiel (Blacktown Mt Druitt Hospital), Adam Bartlett (SCHN) and Gail Matthews (St Vincent's Hospital). Site co-ordinators – Dmitrii Shek (Blacktown Mt Druitt Hospital); Katerina Mitsakos (RNSH); Renier Lagunday (POWH); Sharon Robinson (St George Hospital); Lenae Terrill (Northern Beaches Hospital); Neela Joshi, Ying Li and Satinder Gill (Westmead Hospital); Rebecca Hickey and Alison Sevehon (St Vincent's Hospital). We also wish to thank and acknowledge the use of the University of Wollongong Cryogenic Electron Microscopy Facility at Molecular Horizons, the use of the Victor Chang Cardiac Research Institute Innovation Centre, funded by the NSW Government, and the Electron Microscope Unit at UNSW Sydney, funded in part by the NSW Government.

## Author contributions

R.R., J.Y.H., R.A.B., D.C. and designed the project; J.Y.H. and H.B. performed human B-cell sorting; R.R. generated SARS-CoV-2 proteins and amplified VH and VL from single B-cells; M.S., S.H.J.B. and A.G.S. performed cryo-EM; D.B.L. performed X-ray crystallography; M.D.J and P.M.H. performed animal experiments; G.W., S.G.T. and W.D.R. performed neutralization experiments; R.R., J.Y.H., H. L., J.J., S.U., O.M. and P.S. performed antibody production and BLI experiments; R.R., J.Y.H., D.B.L. and D.C. wrote the manuscript with input from D.L.B., S.H.J.B., M.M., B.H., N.G., J.J.P., A.K., H.M.J., C.C.G. and all other authors.

## Competing interests

The authors declare no competing interests.

## Additional information

[1]Garvan Institute of Medical Research, Sydney, NSW, Australia. [2]UNSW Sydney, St Vincent's Clinical School, Faculty of Medicine, Sydney, NSW, Australia. [3]Center for Inflammation, Centenary Institute and University of Technology Sydney, Sydney, NSW, Australia. [4]Victor Chang Cardiac Research Institute, Sydney, NSW, Australia. [5]UNSW Sydney, School of Medical Sciences, Faculty of Medicine, Sydney, NSW, Australia. [6]Kirby Institute, UNSW Sydney, Sydney, NSW, Australia. [7]Prince of Wales Hospital, Sydney, NSW, Australia. [8]Molecular Horizons, University of Wollongong, Wollongong, NSW, Australia. [9]Royal North Shore Hospital, Sydney, NSW, Australia. [10]Westmead Hospital, Sydney, NSW, Australia. [11]Division of Molecular Immunology, Friedrich-Alexander University Erlangen-Nürnberg and University Hospital Erlangen, Erlangen-Nürnberg, Germany. [12]These authors contributed equally: Romain Rouet, Jake Y. Henry, Matt D. Johansen, Meghna Sobti. [13]These authors jointly supervised this work: Rowena A. Bull, Alastair G. Stewart, Philip M. Hansbro, Daniel Christ. ✉e-mail: r.rouet@garvan.org.au; d.christ@garvan.org.au

