## [Peer Review File · Nature Communications]

Reviewer comments, first round

Reviewer #1 (Remarks to the Author):

In the manuscript by Rouet and colleagues, the authors describe the isolation and characterization of SARS-CoV-2 antibodies that broadly neutralize several variants of concern and display therapeutic properties in vivo. Key to the discovery of these antibodies was an isolation strategy that utilized epitope-specific baits to sort donor B cells, which the authors detail and present as a blueprint for new antibody discovery. The authors then characterize several of their antibodies using binding and structural studies, revealing targeting of the RBD to newly-described class 5 and class 6 epitopes. Overall, the experimental design, execution, and presentation of this work is well executed and warrants publication in this journal. Below I list a couple of minor points the authors may wish to address in a revised manuscript:

For in vivo experiments with K18-mice transgenic for human ACE2, can the authors comment on why an infectious dose of 10^3 PFU was used rather than a 10^4 as seen in most studies (for example Martinez D, 2021, Sci Tran Med)? In addition, I would expect to see higher titers 6 dpi in the vehicle group. Could this be a result of using a lower infectious dose? Given that the limit of detection in the assay is only one log lower than vehicle titers, it is hard to compare the efficacy of the antibodies tested to other SARS-CoV-2 antibody treatments.

The authors suggest that since antibodies GAR05, GAR12 and GAR20 target non-overlapping epitopes, such antibodies may be used in combination therapy. However, they also speculate that mechanistically, one or more of the antibodies could alter the conformation of the spike trimer. The authors could strengthen their claims if they were to provide in vitro evidence that these antibodies could bind together and work synergistically to neutralize the virus.

Given the higher resolution of the X-ray structures detailing Fab-RBD interactions, could the authors speculate on why these antibodies are able to withstand RBD mutations found in Omicron lineages? Could the authors provide a more detailed illustration (either figure or table) of the Hydrogen bonding network, especially at RBD residues that are mutated?

In Figure 3E, I believe this should be labeled i-iii-NTD/RBD, not all labeled as NTD. Also, what does CHO mean? Is this referring to the N343-glycan?

Reviewer #2 (Remarks to the Author):

In this manuscript, the authors use a clever sorting scheme to sort B cells with anticipated epitope specificities. The authors confirm these specificities and identify monoclonal antibodies against a novel "class 6" epitope, which demonstrate broadly binding activity. The manuscript is well written, progresses logically, and conclusions are appropriate. The manuscript could benefit from a few more clarifications (listed below).

Major Concerns

1. Please include a table of monoclonal antibody features, including donor information, gene usage (not just VH and VK/VL genes), CDR3 lengths, and mutations.
2. 30 mg/kg is a very high dose of antibody. Did the authors perform dose titrations? If not, can the authors provide a reference that justifies this high of a dose? It's hard to say these are potentially protective at this dose.

Minor Concerns

1. When referring to therapies in the clinic (real-world scenarios), the authors should use the term effectiveness rather than efficacy to describe the benefit of these measures.
2. At line 100-102, please include the general date range that these samples were collected.
3. Figure 1C, is GAR03 actually neutralizing? It's not standard to have a greater than sign (>) in

front of the concentration and for it to still be considered neutralizing. A precise IC50 is encouraged.

4. What was the resolution for cryo-EM in Fig. 2?

5. For the epitope binning in Figure 1, do the class 6 mAbs compete for RBD binding?

6. Figure 5B – can you please include the WT (Wuhan) neutralization titers in this table?

Reviewer #3 (Remarks to the Author):

In this manuscript by Rouet et al, the authors describe identification of novel broadly neutralizing antibody that recognizes a novel epitope that that classify as Class VI from a convalescent patient infected by ancestral strain (Wuhan) SARS-CoV-2 using epitope-specific B cell sorting together with X-ray/Cryo-EM, live virus neutralization and in vivo efficacy in animal models. The study is well conducted, and inferences supported by data. I have following comments that the authors for authors to consider:

Major:

1. Given the new class of nAb was isolated from a convalescent patient and bnAbs with Omicron neutralizing potential can be generated by immunizations (N Engl J Med 2022; 386:599-601), the need for developing recombinant nAbs that are expensive as therapy need a better justification (immunocompromised/cancer patients individuals and aged population) and the narrative in the introduction needs to be improved accordingly.

2. Since Class VI is newly identified in this study, a clear mechanism for mode of action is required.

3. Why did the authors choose 3-day prior to infection as a time point for prophylactic administration of antibodies? Do the authors know if the administered antibodies remain in circulation longer than 48h? Clearly the abs are working but they may be more efficacious if provided 24-12 h before infection.

4. Based on in vitro neutralization profile Supp Figure 5 it would have been better to choose Beta VOC rather than Delta VOC for in vivo efficacy analyses as Beta appears to be more immune-evasive.

Minor:

1. Please add the that for in vivo experiments, therapeutic administration of antibodies were carried out at +1dpi in the figure legends as well as in the figure for easy comprehension.

We thank the reviewers for the detailed comments. Please find our detailed response as below.

Reviewer #1 (Remarks to the Author):

For in vivo experiments with K18-mice transgenic for human ACE2, can the authors comment on why an infectious dose of 10^3 PFU was used rather than a 10^4 as seen in most studies (for example Martinez D, 2021, Sci Tran Med)? In addition, I would expect to see higher titers 6 dpi in the vehicle group. Could this be a result of using a lower infectious dose? Given that the limit of detection in the assay is only one log lower than vehicle titers, it is hard to compare the efficacy of the antibodies tested to other SARS-CoV-2 antibody treatments.

We have performed extensive dose variation experiments with viral loads starting as high as 10^4 . Our experience is that those mice infected with such high viral loads often succumb to disease with limited inflammation due to overburdening of the immune system. We have previously published with a 10^3 PFU dose (see Counoupas and Johansen, 2021 NPJ Vaccine) and there are many other publications that have also used this viral load. In response to the query regarding the viral titre at 6 dpi, it is worth noting that the peak of viral replication is earlier in the infection (~2-3 dpi). We have opted to use a 6 dpi timepoint as we have extensive experience in profiling viral loads at this timepoint, which is clinically significant as this is when mice succumb to disease due to clinical presentation and severe disease features. We have therefore chosen 6 dpi to maximise clinically relevant data outputs. Furthermore, viral loads as reported in our findings are those recovered from a single lung lobe as compared to the entirety of the lungs and so this may also be factored into our findings. This approach was utilized to allowing additional analyses be performed with the remaining lung tissues. With regards to the comment on the LOD for plaque assay quantification, we would like to draw the reviewer's attention to the lack of any infectious virus recovered in any Ab-treated groups which is noticeably different to those mice that received the isotype Ab.

The authors suggest that since antibodies GAR05, GAR12 and GAR20 target non-overlapping epitopes, such antibodies may be used in combination therapy. However, they also speculate that mechanistically, one or more of the antibodies could alter the conformation of the spike trimer. The authors could strengthen their claims if they were to provide in vitro evidence that these antibodies could bind together and work synergistically to neutralize the virus.

We thank the reviewer for the helpful comment and have now revised Supplementary Fig. 9 (panel B), highlighting the co-binding of GAR05, GAR12 and GAR20 to SARS-CoV-2 RBD measured by biolayer interferometry (BLI). Moreover, panel A of the same figure highlights co-binding of GAR12 and GAR20 with S309 (Sotrovimab), outlining the potential of combination therapy with established monoclonal antibody therapeutics.

Updated Supplementary Fig. 9 shown below;

Given the higher resolution of the X-ray structures detailing Fab-RBD interactions, could the authors speculate on why these antibodies are able to withstand RBD mutations found in Omicron lineages? Could the authors provide a more detailed illustration (either figure or table) of the Hydrogen bonding network, especially at RBD residues that are mutated?

We have now included the following expanded discussion and additional figure (Supplementary Fig. 7, shown below) as requested by the reviewer: (from line 203):

“The large surface area buried at the interface is at the upper end of what is commonly observed for antibody/antigen interactions and is likely reflected by the high affinity of GAR05 towards the RBD (540 pm). However, the surface coverage of GAR05 is in close proximity to multiple positions mutated in various omicron lineages (Fig. 2D). Rationalisation of why tight binding of GAR05 is nevertheless observed can be made by superposition of the GAR05 complex with cryo-EM and crystal structures of an ensemble of omicron variants (B.1.1.529, BA.2, BA.4/5) from a range of biophysical circumstances (RBD down, RBD up, RBD up and complexed by ACE2, see Supplementary Fig. 7). Of the constellation of positions mutated in omicron VOCs, eight are likely to interface with GAR05. Four of these (S477N, T478K, E484A and F486V) adorn a loop demonstrating considerable flexibility and which forms one end of the saddle bound by ACE2 (Supplementary Fig. 7, panel B). Some positions are mutated to smaller side chains and thus are unlikely to present a steric clash (E484A, F486V, K417N and Y505H; Supplementary Fig. 7B). Some positions are mutated to larger side chains albeit presenting in a variety of side chain conformations, some of which might accommodate GAR05 binding (Q493R and N501Y; panel F). Ten hydrogen bonds exist between GAR05 and the RBD (panels G-I). Two of these involve the light chain epitope centred on Y505, both of which will likely be lost by omicron mutants Y505H and N501Y (panel G). The remaining 8 are heavy-chain centric, 7 involved in contacts with conserved RBD targets (both main-chain and side-chain), and only one will be lost by the E484A mutation (panel I). Hence, the ability of GAR05 to maintain tight binding to omicron VOCs appears due to combinatorial effects of; coverage of a large epitope surface, the redundancy of heavy and light chains effectively targeting different surfaces (heavy chain targeting the RBD saddle feature, and the light chain targeting the adjacent Y505 feature (panel A)), the apparent flexibility noted in part of the RBD heavy-chain epitope, the lack of any obvious side-chain mutation likely to present as an unavoidable steric block, and the bulk of hydrogen bonds targeting conserved features.”

And with reference to the indifference of GAR03 to omicron VOCs;... (Line 272) “whereby the bulk of the contact interface is mediated by the antibody light chain, and where the binding epitope is well separated from omicron variant mutation positions (Fig. 3A, positions highlighted in yellow)”.

New Supplementary Fig 7 shown below:

Supplementary Fig. 7. GAR05-RBD superposed with omicron lineage structures. A) GAR05 heavy and light chain (dark and pale green cartoons) bound to Wuhan RBD (cream cartoon) superposed with omicron lineage RBDs (thin red cartoon and sticks). Omicron structures include those from lineages B.1.1.529, BA.2, and BA.4/5, and come from a variety of conformational contexts*. Residues shown as yellow sticks mark positions mutated in omicron lineages. The epitopes recognised by GAR05 heavy and light chains are indicated. The perspective is equivalent to that shown in Fig 2B) Close-up views of GAR05 heavy chain residues in proximity to omicron variant positions. Panel B highlights several mutations (S477N, T478K, E484A and F486V (exclusive to B.A.4/5)), which adorn a relatively flexible loop comprising one end of the saddle to which ACE2 binds. Panel C focuses on the E484A and Q493R positions, whilst panel D highlights the K417N position. Panels E and F highlight light chain residues in proximity to omicron mutations. Panels G-I detail hydrogen bonds (black dashed lines) between GAR05 and the RBD for the light chain (panel G) and the heavy chain (panels H and I). The bulk of these involve RBD side chains or main-chain residues not mutated in omicron variants (RBD features coloured cream), with notable exceptions of the light-chain centric Y505H and N501Y (coloured yellow, panel G) and heavy-chain centric E484A (panel I)

*Omicron structures shown as thin red wire cartoons and sticks comprise; B.1.1.529 in the up and down conformations (PDB 7tgw chains A and B, at 3.0 Å (Ref Ye et al)), B.1.1.529 in the exclusively down conformation (PDB 7wp9, chain A, at 2.56 Å (Ref Yin et al)), B.1.1.529 in the up conformation and complexed with ACE2 (PDB 7wpa chain A, at 2.77 Å (Ref Yin et al)), BA.2 in the exclusively down conformation (PDB 7ub0, chain A, at 3.31 Å (Ref Stalls et al)), BA.4/5 in the exclusively down conformation (PDB 7xnq, chain A, at 3.52 Å (Ref Cao et al)), and BA.4/5 in the up conformation and complexed with ACE2 (PDB 7xwa, chain B, at

3.36 Å (Ref Kimura et al). All structures were determined by cryoEM apart from PDB 7xwa, which was determined by X-ray crystallography.

Ye et al: <https://www.nature.com/articles/s41467-022-28882-9>

Yin et al: <https://www.science.org/doi/10.1126/science.abn8863>

Cao et al: <https://www.nature.com/articles/s41586-022-04980-y>

Stalls et al:

<https://www.sciencedirect.com/science/article/pii/S2211124722007987?via%3Dihub>

Kimura et al:

<https://www.sciencedirect.com/science/article/pii/S0092867422011904?via%3Dihub>

In Figure 3E, I believe this should be labelled i-iii-NTD/RBD, not all labelled as NTD. Also, what does CHO mean? Is this referring to the N343 glycan?

The figure has now been corrected and a specific reference to the N343 glycan included.

Reviewer #2 (Remarks to the Author):

Please include a table of monoclonal antibody features, including donor information, gene usage (not just VH and VK/VL genes), CDR3 lengths, and mutations

We have now included the following Supplementary Table 3. Full sequences for all of the GAR antibodies are also listed in Supplementary Sequences, which directly follow Supplementary Table 3.

Antibody name	VH/VL germlines	VH CDR3 length (Kabat numbering)	Germline mutations
GAR01	IGHV3-30 & IGKV1-33	13	6 in VH, 1 in VL
GAR03	IGHV1-8 & IGLV3-21	12	4 in VH, 4 in VL
GAR04	IGHV1-2 & IGKV4-1	23	2 in VH, 1 in VL
GAR05	IGHV3-66 & IGKV1-33	14	8 in VH, 3 in VL
GAR06	IGHV2-70 & IGLV2-11	12	2 in VH, 4 in VL
GAR07	IGHV3-66 & IGKV1-33	9	4 in VH, 1 in VL
GAR09	IGHV3-66 & IGKV1-33	8	3 in VH, 2 in VL
GAR11	IGHV1-69 & IGLV2-14	17	10 in VH, 2 in VL
GAR12	IGHV3-23 & IGKV1-5	15	6 in VH, 1 in VL
GAR13	IGHV3-13 & IGKV1-39	15	4 in VH, 1 in VL
GAR14	IGHV3-13 & IGKV1-39	17	5 in VH, 4 in VL
GAR15	IGHV3-53 & IGKV1-9	10	3 in VH, 4 in VL
GAR16	IGHV3-13 & IGKV1-39	21	4 in VH, 4 in VL
GAR17	IGHV3-23 & IGLV1-47	17	6 in VH, 5 in VL
GAR18	IGHV4-59 & IGKV3-11	15	4 in VH, 2 in VL
GAR20	IGHV3-34 & IGLV3-21	12	6 in VH, 3 in VL

“...30 mg/kg is a very high dose of antibody. Did the authors perform dose titrations? If not, can the authors provide a reference that justifies this high of a dose? It’s hard to say these are potentially protective at this dose.”

We thank the reviewer for his/her helpful comments. We did not perform dose titrations due to the requirement of carrying out animal work under class 3 biosecurity conditions (which severely restricts throughput). However, the use of 30 mg/kg is well within a range commonly used in the field. This is highlighted by seminal studies by Regeneron Pharmaceuticals whereby dosages of up to 50 mg/kg were employed(<https://doi.org/10.1126/science.abe2402>).

When referring to therapies in the clinic (real-world scenarios), the authors should use the term effectiveness rather than efficacy to describe the benefit of these measures.

This has been corrected.

At line 100-102, please include the general date range that these samples were collected

The following details have been added to the text, “All patients were diagnosed in March 2020 by RT-PCR and follow-up samples collected between May and November 2020”.

Figure 1C, is GAR03 actually neutralizing? It’s not standard to have a greater than sign (>) in front of the concentration and for it to still be considered neutralizing. A precise IC50 is encouraged.

As shown in Figure 1D (middle panel), GAR03 (black line) neutralises SARS-CoV-2 (Wuhan) albeit weakly, precluding the construction of an IC50 curve from the observed range of concentrations. As such, a neutralising IC50 of >20000 ng/ml is presented to denote weakly neutralising activity as opposed to antibodies that have no neutralising capacity at any tested concentration (denoted ‘nn’ in Fig.1C).

What was the resolution for cryo-EM in Fig 2?

The resolution was 3.27 Å, while the resolution for the cryo-EM model presented in Fig 3 was 3.39 Å. These had been detailed in Supplementary Table 2, but have now also been added to the legends of Fig. 2 and Fig. 3.

For the epitope binning in Figure 1, do the class 6 mAbs compete for RBD binding?

Yes, GAR03 and GAR12 compete with each other for RBD binding. Supplementary Fig 9b has been amended to demonstrate RBD binding competition between Class 6 mAbs.

Figure 5B – can you please include the WT (Wuhan) neutralization titers in this table?

The figure has now been amended to clarify A.2.2 (Wuhan) neutralisation titres;

B

IC ₅₀ (ng/ml)	 GAR05	 GAR12	 S309
Wuhan (A.2.2)	82.45	12.77	132.6
BA1	71.31	16.17	804.6
BA2	198.9	26.58	nn
BA5	337.6	62.59	3826

Reviewer #3 (Remarks to the Author):

Given the new class of nAb was isolated from a convalescent patient and bnAbs with Omicron neutralizing potential can be generated by immunizations (N Engl J Med 2022; 386:599-601), the need for developing recombinant nAbs that are expensive as therapy need a better justification (immunocompromised/cancer patients individuals and aged population) and the narrative in the introduction needs to be improved accordingly.

We have now expanded the discussion in the introduction, and specifically refer to immunocompromised populations and the elderly.

2. Since Class VI is newly identified in this study, a clear mechanism for mode of action is required.

We thank the reviewer for his/her helpful comment. We have now extended the discussion, include an additional supplementary figure (Supplementary Fig. 11), and highlight observations that binders to this epitope region disrupt the trimer interface.

“While the class VI antibodies identified here do not directly occlude binding to the cellular ACE2 receptor, cryo-EM experiments performed on GAR03 indicated rapid disruption of the spike structure upon incubation with antibody, suggesting trimer dissociation as a dominant mode of action (similar to what has been observed for the non-ACE2-blocking antibodies CR3022 (a class IV binder), S2H97 (a class V binder) and COV2-3434 (targeting the N-terminal domain) Huo et al, Wrobel et al, Starr et al, Suryadevara et al). This conclusion is further supported by a recently reported antibody, 35B5 (Wang et al), which contacts the class VI epitope largely through unconventional framework interactions (involving outer edges of HCDR2, HCDR3, and framework regions FRH1 and FRH3 – rather than the canonical CDR-mediated interactions reported here – Supplementary Fig. 11). Intriguingly, despite its dramatically different binding mode, 35B5 has also been shown to rapidly disrupt the spike trimer, most likely due to steric issues involving N-terminal domain proximity (Wang et al), further supporting the notion that interaction with the class VI epitope results in general spike dissociation, independent of specific interaction details.”

Huo et al:

<https://www.sciencedirect.com/science/article/pii/S1931312820303516?via%3Dihub>

Wrobel et al: <https://www.nature.com/articles/s41467-020-19146-5>

Starr et al: <https://www.nature.com/articles/s41586-021-03807-6>

Suryadevara et al: <https://pubmed.ncbi.nlm.nih.gov/35472136/>

Wang et al: <https://www.nature.com/articles/s41392-022-00954-8>

Additional Supplementary Fig. 11 shown below:

Supplementary Fig. 11. Superposition of GAR03 and GAR12 Fabs with Fv from 35B5. A) GAR12 and GAR03 Fabs (heavy and light chains as light/dark orange/blue transparent surfaces) positioned against the RBD (cream molecular surface and cartoon) whereas the ACE2 interface and overlaid omicron mutation positions are shaded black and yellow, respectively. The 35B5 Fv is shown as dark purple and violet cartoons (for heavy and light chains). B) The 35B5 antibody binds in an unusual manner with considerable contacts involving heavy-chain framework regions outside of the CDRs.

3. Why did the authors choose 3-day prior to infection as a time point for prophylactic administration of antibodies? Do the authors know if the administered antibodies remain in circulation longer than 48h? Clearly the abs are working but they may be more efficacious if provided 24-12 h before infection.

The half-life of human IgG1 in WT mice is well documented at around 10 days (<https://doi.org/10.1093/intimm/dx1110>). Prophylactic application of antibody at -3 dpi is therefore likely suitable, a decision supported by our results demonstrating viral clearance (Fig. 4C/G).

4. Based on in vitro neutralization profile Supp Figure 5 it would have been better to choose Beta VOC rather than Delta VOC for in vivo efficacy analyses as Beta appears to be more immune-evasive.

We agree with reviewer that it is highly encouraging that GAR05, GAR12 and GAR20 demonstrate broadly neutralising activity across all tested VOC's, including Delta and Beta VOC. Regarding in vivo challenge studies, the Beta variant had already gone extinct at the time of the study which would have raised ethics, biosafety and relevance issues.

Please add that for in vivo experiments, therapeutic administration of antibodies were carried out at +1dpi in the figure legends as well as in the figure for easy comprehension.

We thank the reviewer for his/her helpful comments. The figure and figure legends have now been amended.

Reviewer comments, second round

Reviewer #1 (Remarks to the Author):

The revised manuscript by Rouet and colleagues is much improved and the additional data addresses this reviewer's concerns. My recommendation is that the manuscript be published without delay. However, please note that in Figure 3, the N343 glycan is still labeled as "CHO."

Reviewer #2 (Remarks to the Author):

The authors have largely addressed my concerns

Figure legends for Figure 4 and 5 are switched around.

If the authors are only testing mAbs at 30 mg/kg and calling them potent is not accurate. Moreover, the paper referenced is in NHPs and hamsters which is quite different than what the authors are doing in this manuscript. Moreover, this manuscript shows essentially no difference between 50 mg/kg and 5 mg/kg in their antibody in hamsters. Therefore, the continued use of the term "potent" for in vivo studies is inaccurate as this was not tested.

Reviewer #3 (Remarks to the Author):

Comments:

1. It is now standard in the field to avoid geographical references to any of the SARS-CoV-2 strains. I suggest authors consider using ancestral strain to describe SARS-CoV-2 Wuhan.

REVIEWERS' COMMENTS – 15/12/2022

Reviewer #1 (Remarks to the Author):

The revised manuscript by Rouet and colleagues is much improved and the additional data addresses this reviewer's concerns. My recommendation is that the manuscript be published without delay. However, please note that in Figure 3, the N343 glycan is still labeled as "CHO."

- As requested, the "CHO" carbohydrate presented in Figure 3 has been re-labelled "343 Glycan".

Reviewer #2 (Remarks to the Author):

The authors have largely addressed my concerns

Figure legends for Figure 4 and 5 are switched around.

- Figure Legends for Figures 4 & 5 have been corrected

If the authors are only testing mAbs at 30 mg/kg and calling them potent is not accurate. Moreover, the paper referenced is in NHPs and hamsters which is quite different than what the authors are doing in this manuscript. Moreover, this manuscript shows essentially no difference between 50 mg/kg and 5 mg/kg in their antibody in hamsters. Therefore, the continued use of the term "potent" for in vivo studies is inaccurate as this was not tested.

- The term "potent" has been removed as a descriptor of antibody efficacy.

Reviewer #3 (Remarks to the Author):

Comments:

1. It is now standard in the field to avoid geographical references to any of the SARS-CoV-2 strains. I suggest authors consider using ancestral strain to describe SARS-CoV-2 Wuhan.

- All references to 'Wuhan' have been removed as the term 'ancestral' has been adopted throughout the paper.